# Flux Pinning Properties of Single-Grain Bulk GdBCO Superconductors Processed by Different Thicknesses of Y123 Liquid Source

**DOI:** 10.3390/mi13050701

**Published:** 2022-04-29

**Authors:** Yufeng Zhang, Ziwei Lou, Penghe Zhang, Chunyan Li, Jiaying Zhang, Xiaojuan Zhang

**Affiliations:** College of Mathematics and Physics, Shanghai University of Electric Power, 1851 Hucheng Ring Road, New Pudong District, Shanghai 201306, China; lzw970220@163.com (Z.L.); zphlunwen@163.com (P.Z.); yan156179@163.com (C.L.); 18861121129@163.com (J.Z.); zhangshiep@126.com (X.Z.)

**Keywords:** GdBCO superconductor bulk, top-seeded melt-texture growth, YBa_2_Cu_3_O_7−δ_ liquid source, critical current

## Abstract

The performance of critical current density of GdBa_2_Cu_3_O_7−δ_ (GdBCO or Gd123) superconductor bulk has an important influence on its practical applications. In this work, four single-domain GdBCO superconductor bulks were successfully processed by the modified top-seeded melt-texture growth method. The addition of a YBa_2_Cu_3_O_7−δ_ (Y123) liquid source with different thicknesses, 0 mm (S0), 3 mm (S3), 5 mm (S5), 7 mm (S7), was introduced to study the influence on the superconducting properties. GdBCO bulk with the addition of the Y123 liquid source with a 3-mm thickness shows the best superconducting properties. The addition of the Y123 liquid source results in a decrease in the Gd^3+^ ion concentration required for Gd123 growth; thus, Gd_2_BaCuO_5_ (Gd211) particles in the liquid source need a larger self-decomposition to diffuse Gd^3+^ ions to Gd123 growth front, which refines the size and leads to a homogenous distribution of the Gd211 particles in the bulks. Thus, the more pinning centers of fined Gd211 particles improve the superconducting properties of GdBCO bulk. With increases in the thickness of Y123 liquid source to 5 mm and 7 mm, high RE^3+^ (Gd^3+^ and Y^3+^) concentration can coarsen Gd211 particles and fuse with Gd211 liquid source. The superconducting properties apparently drop. Therefore, the addition of a Y123 liquid source with a suitable thickness is a positive modification to obtain high-performance GdBCO bulk.

## 1. Introduction

Single-grain REBa_2_Cu_3_O_7−δ_ superconducting bulks (REBCO or RE123, where RE denotes the rare-earth elements, such as Y, Gd, Nd, Sm and Eu, etc.) possess a high critical current density (*J*_C_) and a great ability to trap magnetic flux (17.24 T at 29 K for YBCO [1] and 17.6 T at 26 K for GdBCO [2]), making them attractive for practical applications [3,4,5,6], such as magnetic levitation, axial-gap-type rotating machines, magnetic bearings and flywheels, etc. Compared with the conventional top-seeded melt texture growth (TSMG) method of fabricating single-domain REBCO bulks, the top seeded infiltration and growth (TSIG) process could overcome problems such as shape distortion, shrinkage of the final sample and leakage of the Ba_3_Cu_5_O_8_ liquid phase, etc. [7,8,9,10,11]. The Murakami group successfully fabricated single-grain bulk (Gd, Dy) BCO superconductor bulk with an optimization of liquid phase mass using the TSIG technique. Sufficient infiltration of the liquid source leads to a high homogeneity of the micro-structural features to obtain improvements in superconducting properties. Nevertheless, the TSMG method was widely used to fabricate large, single-grain REBCO superconducting bulks that show an excellent magnetic-flux-trapping ability and great potential for batch production growth [12,13]. Under the inhomogeneous furnace environment, Yu et al. [14] fabricated a successful batch process for YBCO bulks with two floors. Zhou et al. [15,16] processed a GdBCO bulk of 32 mm in diameter through the modified TSMG method, which chooses Y123 as a liquid source. The growth along the a-b plane was allowed due to the Y123 liquid source, which depresses the accumulation of Gd211 particles, so as to dramatically improve the properties of the GdBCO bulk. Compared with Nd or Sm light rare-earth elements, the Gd/Ba substitution range in GdBCO materials [17,18,19] is the lowest, and has no serious impact on *T*_C_, which can contribute to the generation of a second peak effect to improve the application potential of bulk materials in the high field. Therefore, the GdBCO system is the most promising to obtain bulks with a high performance in the air using the SMG method.

In the present work, the liquid source Y123 with different thicknesses was introduced to the TSMG process to increase single-domain GdBCO bulk. From the results on magnetic properties and microstructure, the influence of a Y123 liquid source with different thicknesses on the superconducting properties of a single-grain superconductor GdBCO bulks with a 25-mm diameter fabricated by the modified TSMG process was systematically analyzed.

## 2. Materials and Methods

### 2.1. Growth in GdBCO Single Grains

Four GdBCO single grains with the addition of the Y123 liquid source at different thicknesses, respectively, 0 mm (S0), 3 mm (S3), 5 mm (S5) and 7 mm (S7), were fabricated via the well-established TSMG process.

Commercial powders of Gd123, Gd211, Ag_2_O and Pt were used as a starting material for TSMG growth. The molar ratio of Gd123 and Gd211 powders was 1:0.4. To improve the mechanical properties and reduce the coarsening of Gd211 particles, 10 wt% Ag_2_O and 0.5 wt% Pt were also added. The mixed powders were pressed into pellets with the diameter of 25 mm and a thickness of 12 mm. The pellet of NdBa_2_Cu_3_O_7−δ_ (Nd123) seed crystal with a size of 2 mm × 2 mm × 0.5 mm was placed on the GdBCO precursor. Commercially pure Y123 powders were chosen to press into a pellet as the liquid source, which is under the GdBCO precursor. Then, we placed the entire arrangement onto the Al_2_O_3_ sheet into the box furnace. The Y_2_O_3_ pellet of 3-mm thickness was placed onto the Al_2_O_3_ sheet, avoiding the reaction between the Y123 liquid source and the Al_2_O_3_ plate. The temperature profile was as follows. The sample was heated to 1080 °C within 10 h, and held there for 1 h, before being cooled to 1008 °C with a cooling rate of 0.3 °C h^−1^. Eventually, the temperature was decreased to room temperature over 10 h.

After growth, the annealing process was carried out in flowing oxygen. The sample was first heated to 450 °C in 5 h and held for 40 h; then, the temperature was slowly decreased to 350 °C in 140 h, 300 °C in 30 h. Subsequently, it was cooled down to room temperature. The process has been reported elsewhere [19,20,21].

### 2.2. Characterization of GdBCO Single Grains

For the measurements of trapped magnetic flux density, the bulk samples were cooled down to the liquid nitrogen temperature under a magnetic field of 1.0 T. After 30 min, the applied magnetic field was removed, the trapped magnetic flux density distribution was scanned using the automatic Hall probe. The distance between the probe and the sample surface was 0.5 mm. To clarify the effect of liquid source thickness on the performance of the GdBCO superconductor bulk material, small specimens with a size of 2 mm × 2 mm × 1 mm were cut from different bulk sample positions to measure the superconducting properties. There are two specimens under seed position along the *c*-axis growth direction, labeled as C1 about 1.5 mm below the top seed and C2 about 4.5 mm below the top seed, and the other two specimens were along *ab* plane near the boundary, labeled as B1 about 1.5 mm below the surface and B2 about 4.5 mm below the surface. The magnetic test of the sample was measured by a Physical Property Measurement System (PPMS). The critical current density was calculated based on the Bean model [22]. The microstructure and the elemental distribution of the samples were observed by Scanning Electron Microscope (SEM) with energy-dispersive X-ray spectroscopy (EDS).

## 3. Results

### 3.1. Photographs of GdBCO Single Grains

Figure 1 shows the appearance of a GdBCO superconductor bulk growth with the addition of a Y123 liquid source with different thicknesses: S0, S3, S5 and S7, respectively. It can be seen that the Nd123 crystal seed was intact, no melting phenomenon occurred during the growth in the bulk, and the seed fully exerted the role of guiding growth [7,8]. Four folded growth sectors were clearly observed, with no obvious random nucleation. From the top view, four single-grain GdBCO bulks with different thicknesses of Y123 liquid sources were successfully fabricated by the modified TSMG method. As the thickness of the Y123 liquid source increases, the growth ridges become clearer, and the growth plane is more complete. The final dimensions of the bulks with the addition of the Y123 liquid source were almost maintained at around 22 mm. However, the bulk without the addition of the Y123 liquid source (S0) was clearly smaller than the bulks added with the liquid source, and the final diameter was about 21 mm. In addition, the growth line clearly extends to the bottom of the bulks with the addition of the Y123 liquid source, which shows that the samples were fully grown under the guidance of the Nd123 seed crystal. This showed that the addition of the Y123 liquid source can effectively inhibit the shrinkage of the superconductor bulks during the growth. The RE^3+^ solute diffusion model [23,24,25] is widely accepted for the growth in Gd123 grains; the growth is driven by the difference in Gd211 concentration. The low solubility of the Gd ions [26] limits the growth rate, and, subsequently, the size of the Gd123 grains. With the supply of Y123 liquid source, the increase in Gd ions’ solubility effectively curbs the bulk shrinkage and enlarges the size of the GdBCO bulks to improve the superconducting properties of the GdBCO bulks.

### 3.2. Superconducting Properties

Figure 2 shows the *T_C_* curves positioned at B2 of GdBCO superconductor bulks processed with different Y123 liquid source thicknesses. The onset critical temperature of all the GdBCO bulks (*T_C,onset_*) is around 95 K, which shows the high quality of the superconductor bulk. A transition width of (Δ*T*_C_) is 4.2 K at the S0 specimens without a Y123 liquid source, 1.2 K at the S3 specimens, 3.7 K at the S5 specimens and 4.7 K at the S7 specimens, as shown in Table 1. In REBCO superconductor composites, the substitution of RE^3+^ ions to Ba sites can anticipate the *T*_C_ distribution, and a wider superconducting transition can occur, accompanied by the larger substitution of RE to Ba sites [9,10,11]. With the addition of the Y123 liquid source, Gd^3+^ ions doping to Ba site occur mor slightly, and then are enhanced. Thus, the transition width enlarges and more Gd_1+x_Ba_2−x_Cu_3_O_7−δ_ solid solution phases form with the Y123 liquid source reaching 7 mm, which may act as the pinning center to enhance the performance of the superconductor bulk, or destroy the superconducting properties.

The magnetic field dependence of *J*_C_ of specimens cut from different positions, C1, C2, B1 and B2, of the GdBCO bulk processed with the addition of Y123 liquid source in different thicknesses, S0, S3, S5 and S7, respectively, at 77 K, are shown in Figure 3. The *J*_C_–*μ*_0_*H* curves positioned C2 of S7 bulk are excluded in the following analysis, owing to the appearance of the destroyed superconducting properties. The more Gd_1+x_Ba_2−x_Cu_3_O_7−δ_ solid-solution phase may be responsible for this phenomenon, which will be interpreted later. To clarify the result under the low magnetic field, the *J*_C_ in the self field, as a function of the thickness of Y123 liquid source, are plotted in Figure 4. During the growth in the GdBCO bulk, the different *J*_C_ values of specimens from different positions are reported in the previous lectures [10,15], due to the formation of sub-grain boundaries, the distribution of Gd211second-phase particles, etc.

For C1 specimens, there is no obvious change with the 3-mm thickness increase in the Y123 liquid source. Nevertheless, we note that *J*_C_ of S5 with the 5-mm thickness of the Y123 liquid source remarkably increases over the whole magnetic field. When the thickness of Y123 liquid source is 7 mm, *J*_C_ decreases in the whole field and coincides with that of S0 and S3 in the high magnetic field. However, this is higher than that of S0 and S3 in the low and intermediate magnetic field. For C2 specimens, *J*_C_ values increase in the low and intermediate magnetic field with the increase in the Y123 thickness. With C1 specimens of S5, there is a notable increase in *J*_C_ values in the appearance of C2 specimens of S5 in the low and intermediate magnetic field. For B1 and B2 specimens, *J*_C_ values exhibit a remarkable enhancement in the low and intermediate field with the Y123 liquid source thickness of 3 mm, which will be useful for the future application of GdBCO superconductor bulks. Then, *J*_C_ values of S5 and S7 rapidly drop below those of S0. Meanwhile, a clear second peak appears in the *J*_C_ dependence of B1, B2 of S3 and C2 of S5, which can be attributed to the presence of chemical compositional fluctuations due to Gd and Ba sites to form Gd_1+x_Ba_2−x_Cu_3_O_7−δ_ solid solution phase [9,18,19,20]. It can be seen that the maximum critical current density value (*J*_max_) is 4.4 × 10^4^ A/cm^2^ of the B2 specimen positioned near the boundary of the GdBCO bulk with the Y123 liquid source of the 3-mm thickness. This is 71.8% higher than that of the same position of S0 without the Y123 liquid source, which is only 2.56 × 10^4^ A/cm^2^. The bulk of Y123 liquid source with 3-mm thickness exhibits the best performance at low and intermediate fields compared with the other bulks, which shows that superconducting properties obtain a great improvement with the addition of a Y123 liquid source at a suitable thickness. However, with the increase in Y123 liquid source at thicknesses of 5 mm and 7 mm, the *J_C_* value of the GdBCO bulks begins to decrease.

Figure 5 shows the *J_C_*-*μ*_0_*H* curves of B2 specimens of S3 of the GdBCO superconductor bulk at different temperatures. It can be observed that, when the temperature is 80 K, the critical current density at the self-field is the lowest at 3.7 × 10^4^ A/cm^2^, and when the temperature is 40 K, the critical current density at self-field is the highest at 25 × 10^4^ A/cm^2^. The lower the temperature, the stronger superconducting properties, which is consistent with the properties of superconducting materials. With the decrease in temperature, the irreversibility field, which is the magnetic field corresponding to the value of *J*_C_ = 100 A/cm^2^, shifts to the higher magnetic field, and a more clearer second peak is observed at the high magnetic field, which indicates the existence of a Gd_1+x_Ba_2−x_Cu_3_O_7−δ_ solid-solution phase [9,10]. The substituted RE/Ba solid-solution phase possesses a lower *T*_C_ and upper critical field *H*_C2_ [27,28]. When the applied magnetic field increases, the weak superconductivity of the substituted RE/Ba solid solution phase is destroyed and becomes normal, resulting in an effective pinning center with the emergence of the second peak in the *J_C_*-*μ*_0_*H* curve. Thus, the irreversibility field shifts towards to a higher applied magnetic field.

### 3.3. The Distribution of the Trapped Magnetic Field

The distribution of the trapped magnetic field (*B*_trap_) of the GdBCO bulk without and with the 7-mm liquid source is shown in Figure 6. A conical and symmetry trapped flux distribution, the multi-ring of a circle in a two-dimensional (2D) view, shows the high performance of the superconducting bulk [16]. The maximum trapped magnetic field value is 0.116 T in the GdBCO bulk with a Y123 liquid source of 7-mm thickness. This is significantly higher than the 0.058 T of the trapped magnetic field in the GdBCO bulk without a Y123 liquid source. Figure 7 shows the trapped field density of GdBCO superconductor bulks as a function of the thickness of the Y123 liquid source. The tendency increases at the beginning and later declines with the increase in the Y123 liquid source thickness. However, compared with the bulk without the addition of a Y123 liquid source, the trapped magnetic field in all the bulks with the addition of a Y123 liquid source increases. It can be seen that the addition of a Y123 liquid source can effectively strengthen the trapped magnetic flux ability of GdBCO superconductor bulk. However, there is an optimal value for the selection of the Y123 liquid source thickness, which needs to be determined. Y123 liquid source of 3-mm thickness possesses the maximum trapped filed, 0.169 T. As we know, *B*_trap_ = *Aμ*_0_*J*_C_*R* [29,30], here, *A* is the constant, *μ*_0_ is the permeability of the vacuum, *J*_C_ is the critical current density and *R* is the radius of the grain. With the addition of the Y123 liquid source, the size of the GdBCO bulk slightly increases from 21 mm in the bulk without Y123 liquid source to 22 mm in the bulks with Y123 liquid source. Thus, the critical current density *J*_C_ will be a main element of the increase in *B*_trap_, which is consistent with the results of the analysis of the critical current. The trapped field of S3 is the biggest in the four GdBCO bulks. The addition of a Y123 liquid source of a suitable thickness is a positive modification to obtain high-performance GdBCO bulk using the TSMG method.

Table 1 gives the test results of the GdBCO bulks, including trapped magnetic field, superconducting transition temperature and critical current density.

### 3.4. Microstructure

In general, Gd211 particles in the suitable size can be used as pinning center to improve the superconducting properties, as well as the Gd_1+x_Ba_2−x_Cu_3_O_7−δ_ solid-solution phase of a suitable size [10,19,31]. Figure 8 shows the microstructure of the B2 specimens positioned near the boundary of GdBCO bulks with the additions of different Y123 liquid source thicknesses. The white particles were identified as the Gd211 particles by EDS. With the addition of a Y123 liquid source of 3 mm thickness, Gd211 particles have a homogeneous distribution and are refined, which will improve the *J*_C_ performance of the B2 specimens of S3. Some black stripes were observed in B2 of S7, which should be the Gd211 liquid source to destroy the superconducting properties, to result in a lower *J*_C_, as shown in Figure 3.

At the same time, we counted the distribution of the Gd211 particles of different sizes and fitted the average size values of the Gd211 particles (AVG) in these GdBCO bulks using the Nano measure 1.2 software, as shown in the right of Figure 8. The average size values of Gd211 particles first decreased with the addition of a Y123 liquid source of 3 mm thickness, and then increased to values above that of the bulk without the addition of Y123 liquid source. Therefore, we suggested that the appropriate thickness Y123 liquid source will refine the size and introduce the homogenous distribution of Gd211 particles to improve the performance of GdBCO bulk. However, an excessively thick Y123 liquid source will result in the coarsening of Gd211 particles to destroy the superconducting properties, reducing *J*_C_ and trapped field, as shown in Figure 3.

### 3.5. Discussion

Based on the peritectic growth mechanism controlled by RE^3+^ ion diffusion [23,24,25], in the process of melt growth, the growth front of the RE123 crystal is communicated by the RE211 particles through the liquid source, whose connection is interface. Meanwhile, there is a concentration difference in RE^3+^ ions between the growth front of RE123 crystal (RE123/L) and RE211 particles in the liquid source, named the diffusion zone for peritectic reaction. The RE211 particles in the liquid source slowly dissolve and dissociate RE^3+^ ions from the interface of the RE211 liquid phase (RE211/L). Driven by the difference in concentration, RE^3+^ ions diffuse to the RE123/L interface through the liquid source and maintain peritectic reaction with Ba^2+^, Cu^3+^ ions, etc., of the liquid source, in order to form an RE123 phase so as to continue the RE123 growth. Thus, the advancement of the solid–liquid interface will maintain. As the RE123 crystal grows, the RE^3+^ ions at the interface are continuously consumed, resulting in a decrease in the RE^3+^ ions concentration, so that the RE^3+^ ions dissolved from the RE211 particles will diffuse to RE123 growth front through the liquid source under the action of concentration differences, and finally realize the continuous growth in RE123 crystals. Therefore, the concentration and dissolution rate of RE^3+^ ions in the liquid source, element proportion of RE^3+^, Ba^2+^, Cu^3+^, and ions at the growth front will have an important impact on the RE123 growth of crystals.

During the growth in GdBCO bulks, the addition of a Y123 liquid source results in an increase in the solubility of RE ions, which improves the growth rate and size of Gd123 grains [26]. If the Y123 liquid source is supported, the size of GdBCO bulks is 22 mm instead of 21 mm in the GdBCO bulk without Y123 liquid source, as shown in Figure 1. In order to achieve the RE^3+^ ion concentration required for Gd123 growth, Gd211 particles in the liquid source need a large degree of self-decomposition to diffuse Gd^3+^ ions to the Gd123 growth front [7]. Therefore, Gd211 particles should exhibit more obvious refinement and homogenous with the addition of a Y123 liquid source with a thickness of 3 mm. According to the trapping/pushing theory of Gd211 particles in liquid phase [32,33,34], there is a critical radius r* for solid Gd211 particles when RE123 grows and is solidified at a certain rate. When the radius of the Gd211 particles is larger than r*, it will be captured by the RE123 growth front and retained in the growth region. If the radius of the particle is smaller than r*, it will be pushed out by the RE123 growth front and will not stay in the growth zone. The growth in GdBCO bulks is from seed to boundary. Thus, more Gd211 particles with a refined size were pushed to the boundary in the slightly homogenous distribution during the growth in S3 bulks, which formed an effective pinning center so as to apparently improve *J*_C_ the of B1 and B2 specimens of S3 bulks, as well as the superconducting properties, in comparison with B1 and B2 specimens of S0 without the addition of a Y123 liquid source, as exhibited in Figure 3 and Figure 4. Previous studies [35,36] have indicated that the doping of other rare-earth elements can reduce the size of Gd211 particles in bulk materials and slow down the trend of Gd211 particle coarsening. The SEM micrograph of Figure 8 indicates the homogenous distribution and the decrease in the average size values of Gd211 particles in B2 specimens of S3, compared with that of B2 specimens of S0 without the Y123 liquid source.

In S3 bulks, more refined Gd211 particles have been pushed from the seed to boundary, resulting in the decrease in *J*_C_ in the seed position, as shown in Figure 3 and Figure 4. In addition, the larger size of Gd211 particles remains under the seed. Figure 9 shows the microstructure of the C2 and C1 specimens positioned under the seed of GdBCO bulk of S3 and the fitted average size values of Gd211 particles. The reduction in the number and size of Gd211 particles was clearly observed, showing a decrease in the *J*_C_ of C1 and C2, compared with that of B2 in Figure 3 and Figure 4.

According to the Ostwald coarsening theory [37,38,39], when RE^3+^ ions dissolved in the liquid source reaches a certain concentration, the Gd211 particles that remain in the liquid source will undergo coarsening process. In this process, the smaller Gd211 particles dissolve and disappear in the liquid source, and the dissolved RE^3+^ ions migrate to the larger Gd211 particles, making the larger Gd211 particles coarsen and fuse with other particles. Thus, with the addition of the Y123 liquid source with thicknesses of 5 mm and 7 mm, more coarsened Gd211 particles remained under the seed owing to the trapped effect, which are not suitable for the pinning center, resulting in a decrease in superconducting properties in C1 and C2 of S5 and S7. In addition, the number of Gd211 particles in the boundary reduce and the size of Gd211 particles in the boundary increases, which also leads to a *J*_C_ decrease. From Figure 3 and Figure 4, the *J*_C_ of B2 of S5 and S7 markedly drops. The average size values of Gd211 particles increase, as shown in Figure 8. Some black stripes were observed in B2 specimens of S7, which should be due to the fusion of RE^3^ ions and larger Gd211 particles in the liquid source.

During the growth in GdBCO bulks, the Nd123 seed will induce the growth in the Gd123 grain. With the addition of Y123 liquid source, the increase in Gd^3+^ ions’ concentration encourages the substitution of Gd^3+^ to Ba^2+^ sites, resulting in the formation of a Gd_1+x_Ba_2−x_Cu_3_O_7−δ_ solid-solution phase, which possesses a lower *T*_C_ and upper critical field *H*_C2_. A small amount of Gd_1+x_Ba_2−x_Cu_3_O_7−δ_ solid-solution phase causes the appearance of second phase with critical current density under the intermediate magnetic field, as shown in Figure 3 in B1 and B2 specimens of S3 bulks [9,27], which should be useful for practical applications. In addition, this leads to the small transition width of *T*_C_ in B2 of S3, as shown in Figure 2. With the addition of a Y123 liquid source in thicknesses ranging from 5 mm and 7 mm, the increase in Gd^3+^ ions accelerates the substitution of Gd^3+^ to Ba^2+^ site, resulting in the formation of more Gd_1+x_Ba_2−x_Cu_3_O_7−δ_ solid-solution phases, which reduces the *T*_C_ of materials and seriously affects the superconductivity of bulk materials. For the B2 of S5 and S7 in Figure 2, the transition width increases from 3.7 K to 4.7 K. Thus, in Figure 3, the *J*_C_ of S5 and S7 in C1 and C2 positions effectively increases compared with that of S3 in C1 and C2 positions. There is a clear second phase effect in the C2 of S5. There is no curve in Figure 3 for the C2 of S7 owing to the decline in its superconducting properties.

Figure 10 exhibits an SEM microstructure at a magnification (3000×) of C2 specimens of S5 and S7 of GdBCO bulks. The stoichiometry of the various spots labeled in Figure 10 was identified by EDS analysis, and the results are listed in Table 2. Due to the complexity of the microstructure of superconducting materials, EDX results can only provide a possible component analysis. It can be observed that C2 specimens of S5 and S7 are composed not only of Gd123 grains (spot G in C2 specimen of S5) and Gd211 particles (spot D and spot F in C2 specimen of S5), but also Gd211 particles and the BaCuO_2_ and CuO liquid phase (spot B in C2 specimen of S5, spot I, spot K and spot O in C2 specimen of S7), which suggests the appearance of black strips in specimen B2 of S7 in Figure 8. The Gd_1+x_Ba_2−x_Cu_3_O_7−δ_ solid-solution phase also exists in the specimens (spot A, spot C and spot E in C2 specimens of S5, spot J, spot M and spot N in C2 of S7). The following reactions should occur during the growth in GdBCO superconductor bulks.
Gd^3+^+ GdBa_2_Cu_3_O_7−δ_→ Gd_1+x_Ba_2−x_Cu_3_O_7−δ_
2GdBa_2_Cu_3_O_7_ ⇌ Gd_2_BaCuO + 3BaCuO_2_ + 2CuO

With the addition of the Y123 liquid source, the increase in Gd^3+^ ions solution reacts with GdBa_2_Cu_3_O_7−δ_ to accelerate the doping of Gd^3+^ to Ba sites, and, furthermore, form Gd_1+x_Ba_2−x_Cu_3_O_7−δ_, which affects the superconducting properties. Spot L shows Y^3+^ around Gd211 liquid source, which may cause fewer Gd211 particles to be pinned in center to decrease *J*_C_ in B2 of S3, as shown in Figure 3, or destroy the superconducting properties in the C2 of S7, with no appearance of the *J*_C_ curve in Figure 3.

In order to analyze the flux-pinning mechanism of the GdBCO bulk superconductors, the normalized pinning force density, F_p_/F_p,max_, as a function of reduced field, H/H_irr_, of the specimen at C2 position of the GdBCO bulk superconductor of S5, is given in Figure 11. According to Dew-Hughes [40], the peak position of plots, h_0_, can provide information about the pinning mechanism of superconductors [28,41], with h_0_ ≤ 0.3 meaning the normal core pinning (δl pinning) and h_0_ ≥ 0.5 meaning the ∆κ core pinning (δT_c_ pinning). The type δT_c_ pinning is caused by the fluctuations in chemical composition in the superconducting matrix, that is, the weak superconducting region in the matrix, such as the oxygen-deficient region caused by insufficient oxygen infiltration, or the solid-solution phase generated by RE/Ba substitution. In this study, for the C2 specimen under the seed of the GdBCO bulk superconductors of S5, the peak position h_0_ is around 0.45, which indicates a combination of δl pinning and dominant δT_c_ pinning in the C2 specimen positioned under the seed of S5. The bulks exhibit stronger δT_c_ pinning with the increase in Y123 liquid source thickness, and we infer that there are two reasons for this phenomenon: one is that the release of Gd^3+^ ions intensifies the Gd/Ba substitution to form a Gd_1+x_Ba_2−x_Cu_3_O_7−δ_ solid-solution phase, which can cause stronger δT_c_ pinning, improving the superconducting properties, *J*_C,_ as well as the trapped field. The other is that more RE^3+^ ions can contribute to the coarsening of Gd211 particles, which can lead to fewer Gd211 particles serving as effective pinning centers and weaker δl pinning. The analysis of pinning mechanism can prove the previous conclusion.

## 4. Conclusions

Four single-domain GdBCO superconductor bulks with a diameter of 25 mm and a thickness of 12 mm were successfully fabricated by the modified TSMG technique. The superconducting properties of the bulk with the addition of the Y123 liquid source at different thicknesses of 0 mm (S0), 3 mm (S3), 5 mm (S5) and 7 mm (S7) were compared. By studying the magnetic flux field, critical transition temperature and critical current density of samples, we found that the superconducting properties of the bulks are greatly improved with the addition of the Y123 liquid source. When the Y123 liquid source thickness is 3 mm, GdBCO bulk exhibits the best superconducting properties. The maximum magnetic flux field of the sample is 0.169 T, and the highest critical current density is 4.4 × 10^4^ A/cm^2^ at 77 K in self field, 71.8% higher than that of the same position of GdBCO bulk without the addition of the Y123 liquid source. The onset critical temperature of all the GdBCO bulks is around 95 K, with superior superconducting properties. According to the peritectic growth mechanism controlled by RE^3+^ ion diffusion, the addition of a Y123 liquid source results in the increase in the solubility of RE ions, which improves the growth rate and then the size of Gd123 grains. When the Y123 liquid source is supported, the size of GdBCO bulks is 22 mm instead of 21 mm in the GdBCO bulk, without a Y123 liquid source. To achieve the Gd^3+^ ion concentration required for Gd123 growth, Gd211 particles in the liquid source require a large degree of self-decomposition to diffuse Gd^3+^ ions to the Gd123 growth front. Therefore, Gd211 particles should exhibit a more obvious refinement and become homogenous with the addition of a Y123 liquid source at a thickness of 3 mm, which enhances the *J*_C_ and trapped field of GdBCO bulks. With the addition of a Y123 liquid source with thicknesses of 5 mm and 7 mm, more RE^3^ ions result in more coarsened Gd211 particles remaining and fusing with the Gd211 liquid source, which are not suitable to be the pinning center, reducing the superconducting properties of S5 and S7. EDS element analysis supports our suggestion. The release of Gd^3+^ intensifies the Gd/Ba substitution to form a Gd_1+x_Ba_2−x_Cu_3_O_7−δ_ solid-solution phase, which can cause a stronger δT_c_ pinning and the appearance of second phase in critical current density, to improve the superconducting properties, *J*_C,_ as well as the trapped field. During the modified TSMG process, there is an optimal thickness for the addition of the Y123 liquid source based on the size of the GdBCO bulk.

## Figures and Tables

**Figure 1 micromachines-13-00701-f001:**
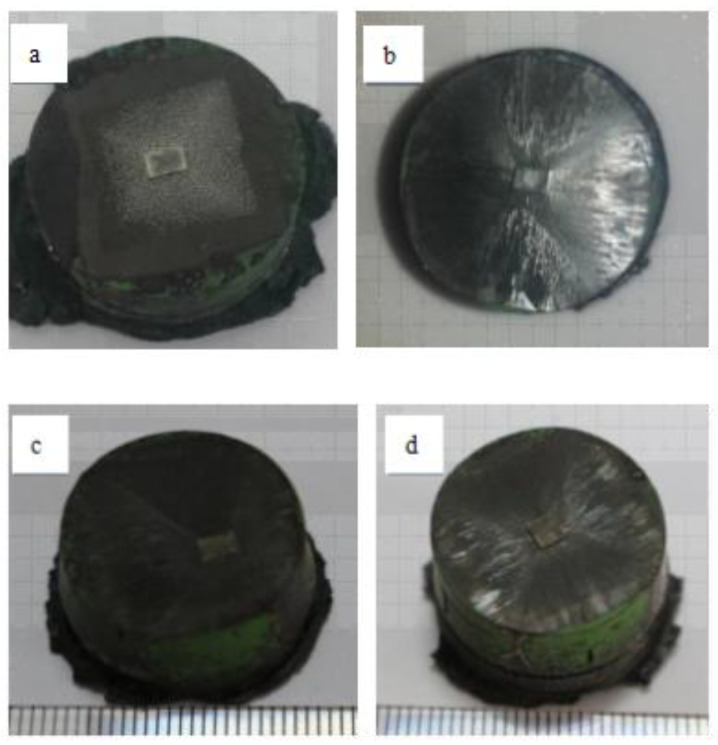
Photographs of GdBCO single grains prepared by TSMG process with additional different thicknesses of Y123 liquid source: (**a**) 0 mm (S0); (**b**) 3 mm (S3); (**c**) 5 mm (S5); (**d**) 7 mm (S7); (**e**) assembly drawing.

**Figure 2 micromachines-13-00701-f002:**
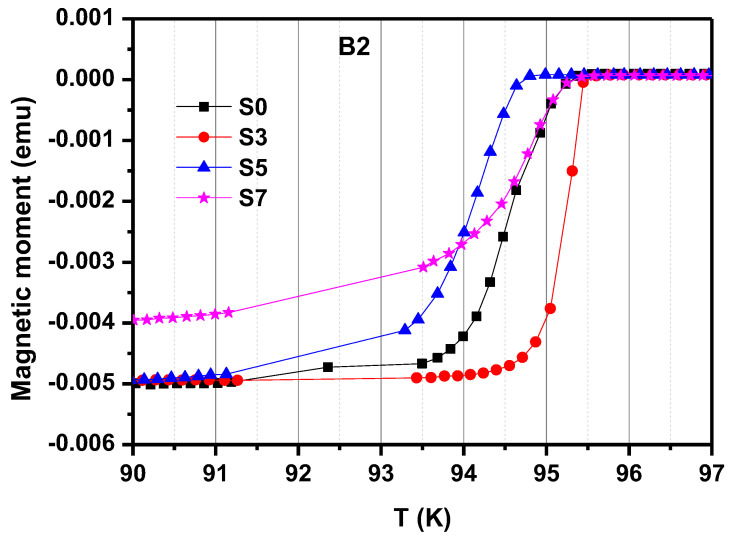
Superconducting transition temperature (*T_C_*) curves at position B2 of the GdBCO bulks processed with liquid source in additional different thicknesses of 0 mm (S0), 3 mm (S3), 5 mm (S5), 7 mm (S7).

**Figure 3 micromachines-13-00701-f003:**
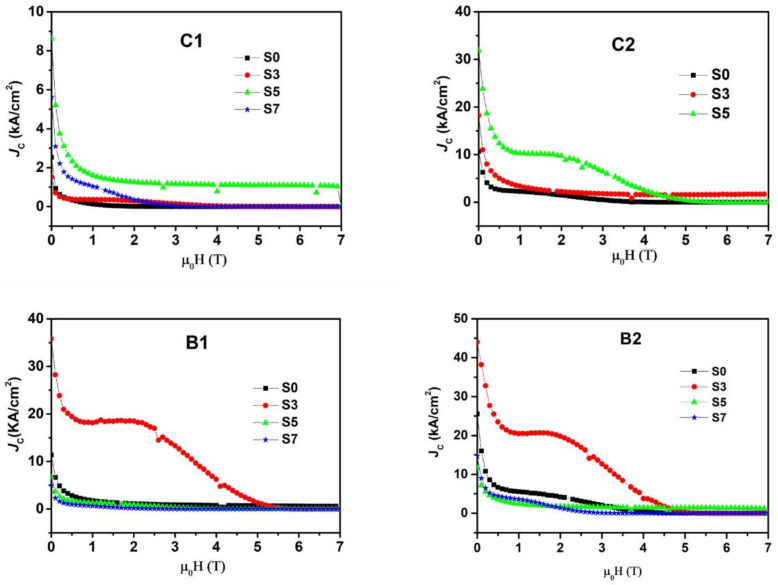
The *J_C_*-B curves of the samples in the different positions of the GdBCO bulk processed with liquid source at different additional thicknesses of 0 mm (S0), 3 mm (S3), 5 mm (S5), 7 mm (S7) at 77 K.

**Figure 4 micromachines-13-00701-f004:**
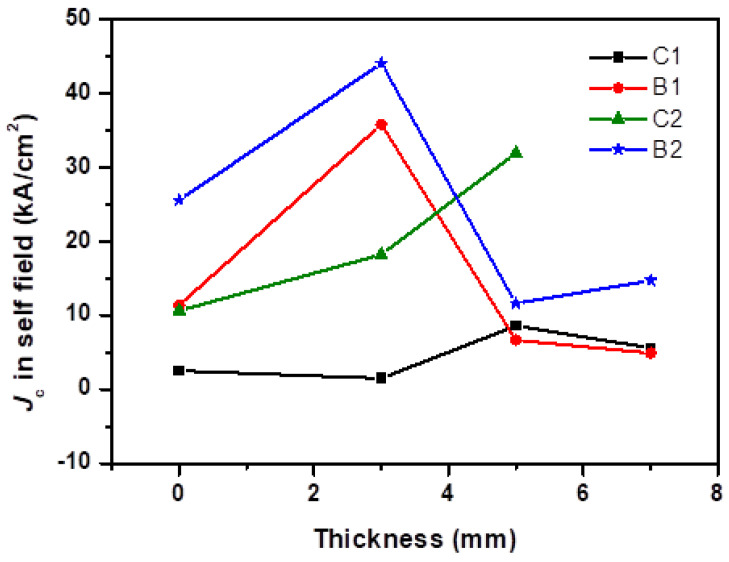
Plots of *J_C_* in self field at position B1, B2, C1, C2 of the GdBCO bulks processed with liquid source in additional different thicknesses of 0 mm (S0), 3 mm (S3), 5 mm (S5), 7 mm (S7).

**Figure 5 micromachines-13-00701-f005:**
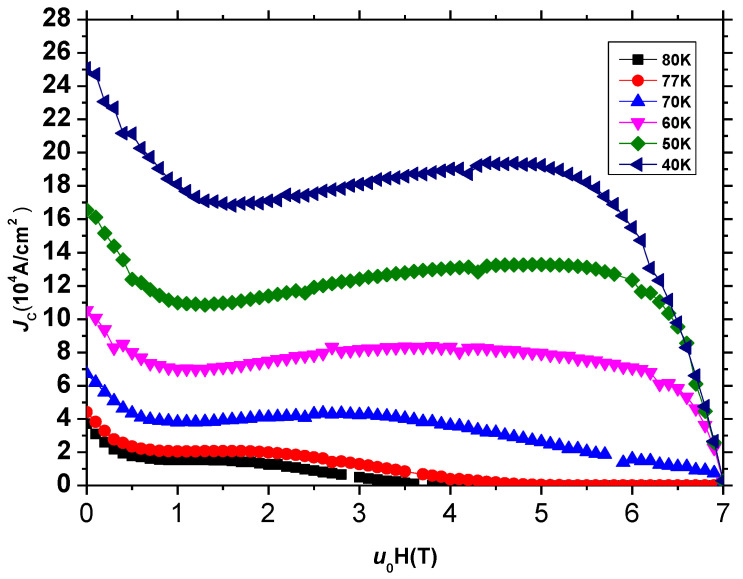
The *J_C_*-B curves of B2 specimen of S3 of GdBCO superconductor bulk at different temperatures.

**Figure 6 micromachines-13-00701-f006:**
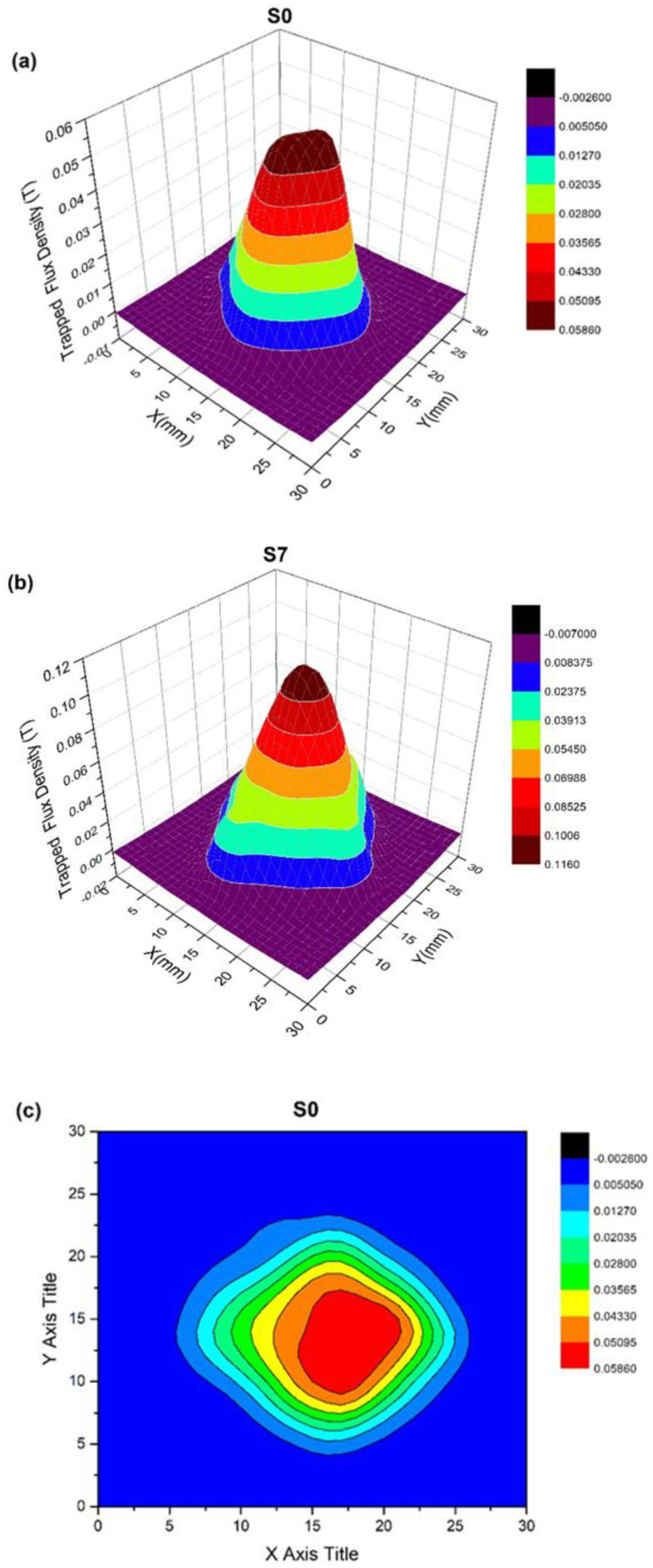
Trapped magnetic flux density profiles of GdBCO bulk with Y123 liquid source in addition to different thicknesses: (**a**) S0; (**b**) S7; (**c**) S0; (**d**) S7.

**Figure 7 micromachines-13-00701-f007:**
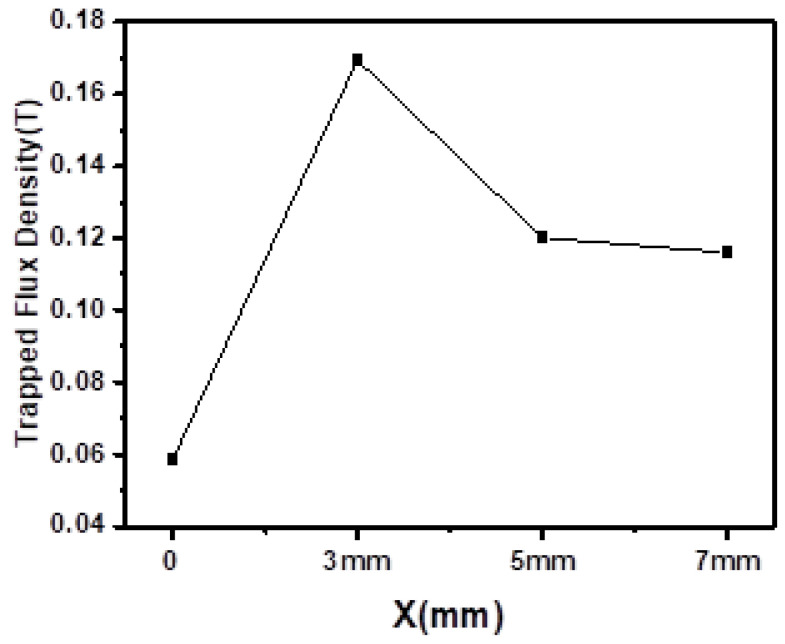
Trapped magnetic flux density profiles of GdBCO bulk with Y123 liquid source in additional different thicknesses.

**Figure 8 micromachines-13-00701-f008:**
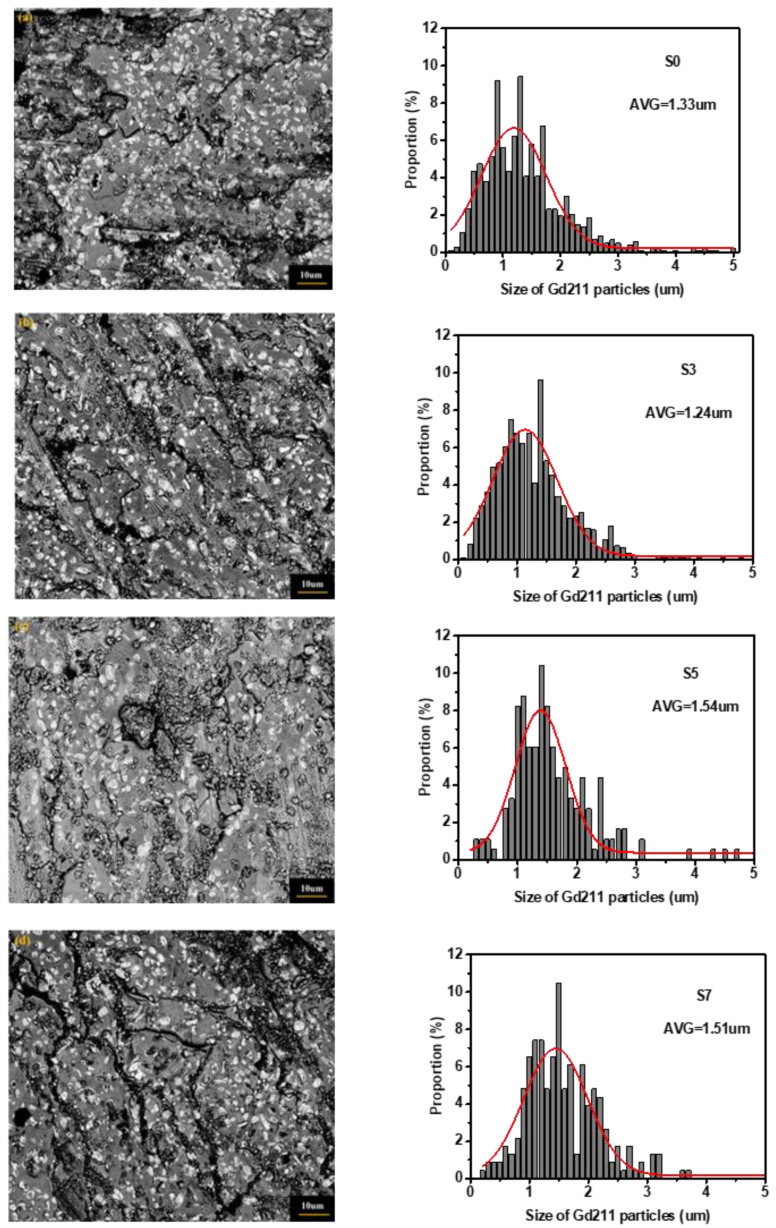
The SEM micrographs imaged with the magnification of 3000× of B2 specimens of GdBCO bulks with the addition of Y123 liquid source of additional, different thicknesses: (**a**) S0; (**b**) S3; (**c**) S5; (**d**) S7, respectively. The corresponding size histograms of Gd211 particles, along with the fitted curves, are given at the corresponding right side of the image, respectively. This also shows the average size values of the Gd211 particles (AVG) from the peak position of the fitted curves.

**Figure 9 micromachines-13-00701-f009:**
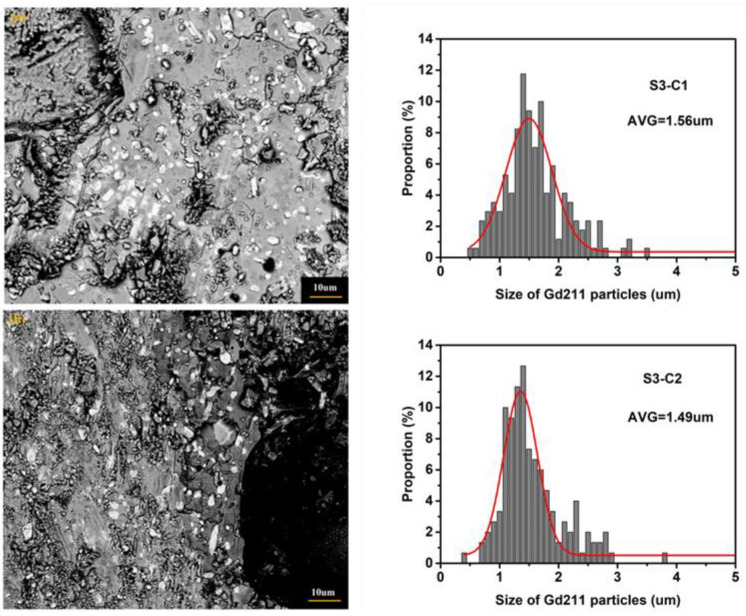
The SEM micrographs imaged with the magnification of 3000× of C1, C2 specimens of GdBCO bulk with Y123 liquid source in the thickness of 3 mm, S3, (**a**) C1; (**b**) C2, respectively. The corresponding size histograms of Gd211 particles and fitting curves are given at the corresponding right side of the image, respectively. This also shows the average size values of the Gd211 particles (AVG) from the peak position of the fitted curves.

**Figure 10 micromachines-13-00701-f010:**
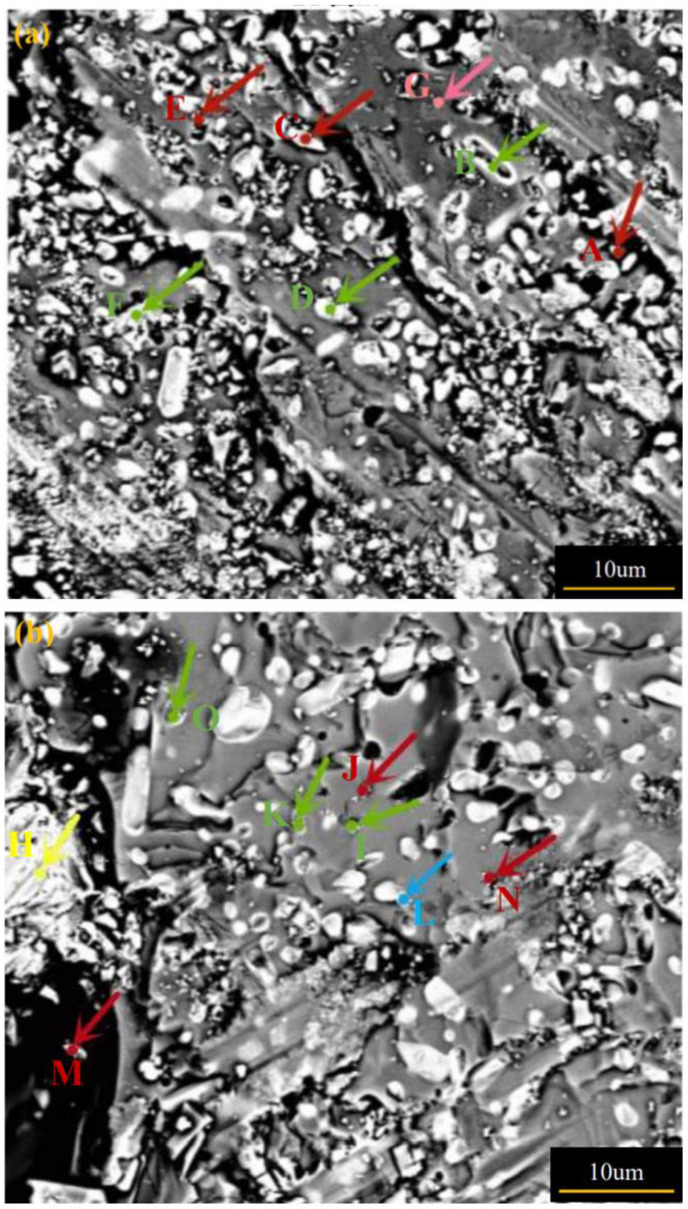
SEM micrographs (at a magnification of 3000×) of specimens at C2 position of the GdBCO superconductor bulks: (**a**) S5, (**b**) S7. The Gd211 particles, Gd123 phase and pure Gd123 particles, Ag particles and spot containing Y^3+^ particles are marked with green dots, red dots, pink dot, yellow dot and blue dot, respectively.

**Figure 11 micromachines-13-00701-f011:**
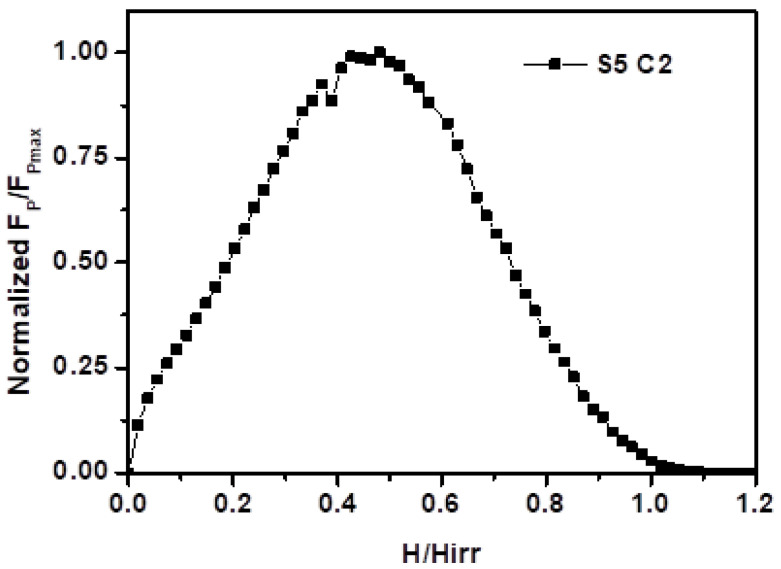
Normalized pinning force density, F_p_/F_p,max_, as a function of reduced field, H/H_irr_, of C2 specimen positioned under the seed of the S5 GdBCO bulk superconductor at a constant temperature of 77 K.

**Table 1 micromachines-13-00701-t001:** The corresponding data obtained from the measurement of GdBCO bulks.

Sample		S0	S3	S5	S7
Δ*T*_C_		4.2 K	1.2 K	3.7 K	4.7 K
*B* _trap_		0.058 T	0.169 T	0.120 T	0.116 T
*J*_C,self-field_(10^4^ A/cm^2^)	C1	0.25	0.15	0.86	0.56
	C2	1.06	1.82	3.19	
	B1	1.14	3.58	0.67	0.49
	B2	2.56	4.4	1.16	1.47

**Table 2 micromachines-13-00701-t002:** Stoichiometry and possible particles of spots labeled identified by EDS analysis.

Sample	Spot	Gd/%	Ba/%	Cu/%	O/%	Y/%	Ag/%	Possible Particles
S5-C2	A	9.8	15.9	24.3	48.4			Gd_1+x_Ba_2−x_Cu_3_O_7−δ_ (Gd:Ba:Cu = 1.21:1.96:3.00)
	B	22.1	15.2	17.2	45.5			Gd_2_BaCuO_5_ + 0.38BaCuO_2_ + 0.18CuO (Gd:Ba:Cu = 2.00:1.38:1.56)
	C	10.8	16	23	49.9			Gd_1+x_Ba_2−x_Cu_3_O_7−δ_ (Gd:Ba:Cu = 1.41:2.09:3.00)
	D	19.7	10.8	11.4	58.2			Gd_2_BaCuO_5_ (Gd:Ba:Cu = 2.00:1.10:1.16)
	E	7.9	12.4	18.6	61.1			Gd_1+x_Ba_2−x_Cu_3_O_7−δ_ (Gd:Ba:Cu = 1.27:2.00:3.00)
	F	21.6	11.5	11.7	55.2			Gd_2_BaCuO_5_ (Gd:Ba:Cu = 2.00:1.06:1.08)
	G	8.5	16.5	24.3	50.6			GdBa_2_Cu_3_O_7−δ_ (Gd:Ba:Cu = 1.05:2.04:3.00)
S7-C2	H						100	Ag
	I	18.1	11.6	13.3				Gd_2_BaCuO_5_ + 0.28BaCuO_2_ + 0.19CuO (Gd:Ba:Cu = 2:1.28:1.47)
	J	7.9	15	23.8				Gd_1+x_Ba_2−x_Cu_3_O_7−δ_ (Gd:Ba:Cu = 1.00:1.89:3.00)
	K	26.2	14.8	17.5				Gd_2_BaCuO_5_ + 0.13BaCuO_2_ + 0.21CuO (Gd:Ba:Cu = 2:1.13:1.34)
	L	18.2	10.5	12.1		1		Y^3+^ + Gd_2_BaCuO_5_ + 0.15BaCuO_2_ + 0.18CuO (Gd:Ba:Cu:Y = 2:1.15:1.33:0.9)
	M	17.6	13	21.9				Gd_1+x_Ba_2−x_Cu_3_O_7−δ_ (Gd:Ba:Cu = 2.41:1.78:3)
	N	11.3	17.2	26.3				Gd_1+x_Ba_2−x_Cu_3_O_7−δ_ (Gd:Ba:Cu = 1.29:1.96:3)
	O	26.9	14.1	15.9				Gd_2_BaCuO_5_ + 0.05BaCuO_2_ + 0.13CuO (Gd:Ba:Cu = 2:1.05:1.18)

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
