# Peer review of "Flux Pinning Properties of Single-Grain Bulk GdBCO Superconductors Processed by Different Thicknesses of Y123 Liquid Source"

_micromachines, 2022, doi:10.3390/mi13050701_

Round 1
Reviewer 1 Report
The work done by the authors is very appreciated. Please do the following
- Please add before second section, the splitting of next sections
- Strictly arrange the paper as per he template by removing the gaps in pages
- Make some comparative assessment in the form of table and compare your results as numerical values including citations, which strengthens the paper more
- Fig 6, the axis values are not readable, make it zoom
- Fig 7, compare the same with few other materials also if possible (Without and with Y123)
- Some abbrevations are abbrevated more times. Please expand only at their 1st usage
- Table 1, why not you add some advantages and limitations? check once
Reviewer 2 Report
The authors mainly studied the influence of the addition of Y123 liquid source with different thicknesses during the growth of GdBCO on the superconducting properties. This work can help people to understand the physical mechanism of Y123 in the growth and provide new opportunities for real applications. There are minors points to be addressed.
Firstly, in the figure1, the absence of some necessary auxiliary lines (such as scale bar etc) makes me hard to get your views and conclusions in your main text. I can’t even distinguish where the Y123 liquid source is or anything about the GdBCO bulk. Thus, I hope you to draw a more elaborate figure. Secondly, why do you only choose specimens of such two heights which at 1.5mm and 4.5mm below the top seed for the measurement of the superconducting properties? In the text, any explanations or cites about this two numbers are lacking. I wonder whether the results would be different while the measuring specimens are located at any other positions? For example, the bulk of Y123 liquid source with 5mm thickness exhibits better performance for C1 and C2 specimens while not for B1 and B2 specimens. So, in my view, this is an important issue that you need to solve. Thirdly, in the part of ‘Superconducting properties’, due to the few numbers of the measuring samples, I think that the results in your text are unprecise. If I were you, I will do more work about different thicknesses around 3mm with higher accuracy
(such as 2.5mm and 3.5mm). If you do more about it, you can do better. For C1 specimens, there is no second peak in the Jc dependence of all samples at 77K which is different from others. It is very strange; can you explain for the difference? And can the second peak appears with the decrease of the temperature? Fourthly, in 3.3 ’ The distribution of the trapped magnetic field’, the trapped magnetic flux density profiles of GdBCO bulk with Y123 liquid source in 3mm and 5mm is absent. If you want to emphasize the significance, I think it is essential. And in this part, I don’t access any helpful information to understand the “distribution” of the trapped magnetic field. You had not described the “distribution” for me, and you just were comparing the trapped field density of GdBCO superconductor bulks with the different thickness of Y123 liquid source. That seems a little off your subtitle. I think this paragraph needs rewriting, and you should focus on the “distribution” of the bulk with 3mm thick Y123 liquid source.
Reviewer 3 Report
The paper “Flux pinning properties of single-grain bulk GdBCO superconductors processed by different thickness of Y123 liquid source” by Y.F. Zhang et al. presents interesting and useful results and is suitable for publication in MDPI "Micromachines" journal. To improve the manuscript, I suggest the following minor corrections:
- The YBCO high-Tc superconductor is usually more common and better known than GdBaCuO. Therefore, for a general reader it would be useful to compare the obtained superconducting properties of GdBaCuO with those of typical YBCO high-Tc superconductors.
- Minor stylistic/language corrections are required. For example, Line 146, “have been report” -> “have been reported”, etc.
